# The Employment of Genera *Vaccinium*, *Citrus*, *Olea,* and *Cynara* Polyphenols for the Reduction of Selected Anti-Cancer Drug Side Effects

**DOI:** 10.3390/nu14081574

**Published:** 2022-04-10

**Authors:** Jessica Maiuolo, Vincenzo Musolino, Micaela Gliozzi, Cristina Carresi, Francesca Oppedisano, Saverio Nucera, Federica Scarano, Miriam Scicchitano, Lorenza Guarnieri, Francesca Bosco, Roberta Macrì, Stefano Ruga, Antonio Cardamone, Anna Rita Coppoletta, Sara Ilari, Annachiara Mollace, Carolina Muscoli, Francesco Cognetti, Vincenzo Mollace

**Affiliations:** 1Laboratoy of Pharmaceutical Biology, IRC-FSH Department of Health Sciences, University “Magna Græcia” of Catanzaro, Campus Universitario di Germaneto, 88100 Canzaro, Italy; v.musolino@unicz.it; 2IRC-FSH Department of Health Sciences, University “Magna Græcia” of Catanzaro, 88100 Catanzaro, Italy; gliozzi@unicz.it (M.G.); carresi@unicz.it (C.C.); oppedisanof@libero.it (F.O.); saverio.nucera@hotmail.it (S.N.); federicascar87@gmail.com (F.S.); miriam.scicchitano@hotmail.it (M.S.); lorenzacz808@gmail.com (L.G.); boscofrancesca.bf@libero.it (F.B.); robertamacri85@gmail.com (R.M.); rugast1@gmail.com (S.R.); tony.c@outlook.it (A.C.); annarita.coppoletta@studenti.unicz.it (A.R.C.); sara.ilari@hotmail.it (S.I.); mollace@libero.it (V.M.); 3Nutramed S.c.a.r.l, Complesso Ninì Barbieri, Roccelletta di Borgia, 88021 Catanzaro, Italy; 4Medical Oncology 1, Regina Elena National Cancer Institute, IRCCS, 00144 Rome, Italy; annachiaramollace@gmail.com (A.M.); fcognetti@gmail.com (F.C.); 5IRCCS San Raffaele, Via di Valcannuta 247, 00133 Rome, Italy

**Keywords:** chemotherapy, genera *Vaccinium*, *Citrus*, *Olea*, *Cynara*, Cisplatin, Doxorubicin, Tamoxifen, Paclitaxel, polyphenols

## Abstract

Cancer is one of the most widespread diseases globally and one of the leading causes of death. Known cancer treatments are chemotherapy, surgery, radiation therapy, targeted hormonal therapy, or a combination of these methods. Antitumor drugs, with different mechanisms, interfere with cancer growth by destroying cancer cells. However, anticancer drugs are dangerous, as they significantly affect both cancer cells and healthy cells. In addition, there may be the onset of systemic side effects perceived and mutagenicity, teratogenicity, and further carcinogenicity. Many polyphenolic extracts, taken on top of common anti-tumor drugs, can participate in the anti-proliferative effect of drugs and significantly reduce the side effects developed. This review aims to discuss the current scientific knowledge of the protective effects of polyphenols of the genera *Vaccinium*, *Citrus*, *Olea*, and *Cynara* on the side effects induced by four known chemotherapy, Cisplatin, Doxorubicin, Tamoxifen, and Paclitaxel. In particular, the summarized data will help to understand whether polyphenols can be used as adjuvants in cancer therapy, although further clinical trials will provide crucial information.

## 1. Introduction

A correct diet, rich in fruits, vegetables, and fibers, with moderate consumption of milk, dairy, meat, and animal products or animal fats plays a protective role in preventing cancer. In particular, the Mediterranean diet, which precisely reflects this nutritional program, is the best food model to reduce the onset of cancer [1,2]. More than 100 natural plant-based compounds possess anti-cancer properties, killing cancer cells without being toxic to healthy cells [3,4]. Most of the molecules with these characteristics belong to the family of polyphenols, a group of natural compounds widely distributed in the plant kingdom; polyphenols are characterized by more than 8000 phenolic structures [5,6]. Polyphenols are made, as the name indicates, from a set of phenolic structures, in which a OH group replaces a benzene hydrogen atom and whose chemical formula is C_6_H_5_OH. These compounds are secondary plant metabolites and possess countless beneficial activities on human health [7]. Among the beneficial properties of polyphenols, it is appropriate to remember the antioxidant, anti-microbial, anti-inflammatory, anti-cancer, and anti-diabetic properties regulating the intra and intercellular signaling pathways [8]. Following the ingestion, polyphenols are absorbed and transformed into bioactive compounds; initially, the enzymatic splitting of the carbohydrate portion (when present) occurs, and then aglycones enter the epithelial cells of the small intestine by passive diffusion. Polyphenolic compounds that cannot be absorbed or large ones reach the colon, where the microbiota will metabolize them [9]. The aglycones and the final derivatives absorbed (in the small intestine or the colon, respectively) are conjugated by methylation, sulfation, and glucuronidation in the enterocytes and then enter the bloodstream through the portal vein. Then, they reach the liver, where they may be subjected to more phase II metabolism; they are subsequently transported to the various tissues or secreted in urine. Finally, the unabsorbed metabolites are eliminated via feces [10]. Figure 1 shows a summary of polyphenol absorption and metabolism. This review is divided into three parts: the first section will deepen the main characteristics of cancer and describe the mechanisms of action by which the most used drugs try to counter the uncontrolled proliferation, a characteristic of this disease. The mentioned anticancer drugs will include alkylating agents, antimetabolites, agents that modulate hormonal synthesis, and antimitotic drugs; more specifically, one drug for each category will be investigated: Cisplatin, Doxorubicin, Tamoxifen, and Paclitaxel, respectively. In the second section of the review, we will deepen some genera of plants with recognized anticancer properties. Since countless compounds of plant origin possess anti-tumor activity, we have arbitrarily chosen four of the most cited genera in the scientific literature: genera *Vaccinium* L., *Citrus* L., *Olea* L., and *Cynara* L. Finally, the third section of the review will be devoted to understanding the mechanisms through which the four chosen genera act by reducing the side effects of the selected anti-cancer drugs.

## 2. Cancer: Pharmacological Treatments and Their Side Effects

Cancer is the second leading cause of death, overcome only by heart disease [11]. Until a hundred years ago, cancer was not as common as it is today, and its incidence has increased alarmingly as a result of several factors such as the change in lifestyle and daily habits and the increase in life expectancy [12]. A tumor is an abnormal growth of cells with impaired function, which, in the most burdened cases, may spread to adjacent cells, organs, or other parts of the body other than the site of origin. In this case, the formation of metastases occurs via lymphatic or haematogenic diffusion through the blood [13]. Generally, there is no real symptomatology prior to the expansion of the tumor mass, while when the mass increases or shifts, symptoms such as weight loss, fatigue, fever, and skin changes can appear. However, it is crucial to highlight that it is impossible to generalize, and each cancer has its characteristics that also depend on the individual subject [14]. Several treatments for cancer depend on the state of advancement, the type of cancer, and the tissue or organ directly involved. The most widely used options are chemotherapy, surgery, radiation therapy, targeted hormonal therapy, or, in most cases, a combination of these treatment methods. Unfortunately, each of the pointed treatments determines substantial side effects for the patient, who must be informed before starting treatment in order to assess, together with the oncologist, the risk–benefit relationship [15]. Chemotherapy is based on antitumor drugs, which, with different mechanisms, interfere with the growth of cancer cells. However, chemotherapy shows several limitations since it is not specific to cancer cells destroying healthy tissues causing their loss [16]. The administration of antitumor drugs involves the onset of different side effects that depend on the type of employed drugs, which limit and/or alter the quality of life. Even worse, the plethora of side effects can limit the success of the chemotherapy program itself due to the need for dosage reduction on one side and patient drop-off on the other [17]; finally, drug resistance can arise, and it no longer fulfills its function. Chemotherapeutic treatments are prescribed in measured quantities and limited time [18]. Sometimes chemotherapy is administered as two or more medications simultaneously; in this way, lower dosages are used, and treatment times can be reduced. Currently, the chemotherapeutic drugs used are divided according to the mechanism of action in alkylating agents, antimetabolites, immunological elements, hormonal components, or antimitotic [19]. The alkylating agents act by intercalating the alkyl groups between the double helix of DNA. Alkylating agents in this way prevent DNA replication and induce an alteration in RNA transcription. By blocking these systems, the cell is no longer able to complete protein synthesis and encounters the mechanism of programmed cell death called apoptosis. According to the fundamental mechanism of action, they are classified into three groups: platinum-based agents (Cisplatin, Carboplatin, Oxaliplatin), organophosphorus compounds, and nitrogen mustards (Cyclophosphamide, Chlorambucil) [20]. Antimetabolites interfere with the formation and/or use of a normal metabolite present within the cell. In particular, these agents inhibit the synthesis of de novo DNA, having a chemical structure similar to purines or pyrimidines that can easily be substituted. They also affect the activity of many crucial enzymes such as dehydrogenases, topoisomerases, nucleosides, and kinases [21]. The employment of endocrine therapy occurs when patients present cancer cells with hormone-positive receptors. This approach can employ the antagonist of the estrogenic receptor (i.e., tamoxifen) class of drugs and aromatase inhibitors (i.e., trozole or exemestane) [22]. Finally, the antimitotic agents induce the reduction of proliferation and the invasion of cancer cells thanks to the alteration of the function of cytoskeleton microtubules, which causes cell cycle arrest and subsequent apoptotic death [23].

### 2.1. The Alkylating Agent Cisplatin

Cisplatin is one of the most well-known metal-based chemotherapeutic drugs, and, to date, it is used to treat 50% of all patients. Although it was discovered in 1845, its mechanism of action, based on the inhibition of cell division, was described only in 1965 [24]. It is used to treat a variety of solid cancers such as ovarian, testicular, lung, gastric, bladder, cervical, head, and some other cancers. Cisplatin is administered intravenously to the patients with a sterile saline solution [25]. Because of the high chloride concentration (100 mM), the drug remains intact, and it may covalently associate with proteins present in the serum, particularly with albumin. Entry into cells, both tumors and healthy cells, occurs mainly by passive diffusion; the uptake can also be mediated, with an active mechanism, from the trans-membrane protein CRT1, responsible for the transport of copper [26]. Within cells, the chloride concentration is much lower (4–20 mM), and hydrolysis reactions occur; one or both chlorides of cisplatin are released, forming very reactive electrophilic groups, which strongly bind with various nucleophiles [27]. The mono or dihydrate platin enters the nucleus and reacts with the nucleophilic groups on each DNA base [28]. The chemotherapy drug can bind both genomic and mitochondrial DNA through this mechanism, causing injury, stopping the synthesis, transcription, and translation of the macromolecule, and activating different transduction pathways that lead to apoptosis or necrosis [29]. In addition, the drug can activate proteins with the function of damaging the cell; high-mobility group proteins, such as high-mobility group box protein 1 (HMGB1), recognize cisplatin-DNA adducts and bind to it, shielding the DNA from repairing. HMGB1 could protect cisplatin-DNA adducts from DNA repair enzyme recognition [30]. Another mechanism used by cisplatin to induce the mortality of cancer cells is to accumulate reactive oxygen species (ROS), which lead to lipid peroxidation, depletion of sulfhydryl groups, alteration of different transduction pathways, and apoptotic death. Cisplatin can also induce cellular apoptosis by stopping the cell cycle in the stages prior to mitosis (G1, S, and G2) and inducing protein p53 [31]. Unfortunately, the side effects generated by cisplatin are multiple, including hepatotoxicity, nephrotoxicity, ototoxicity, gastrointestinal toxicity, and neurotoxicity [32,33,34]. It is appropriate to add the induction of the drug-resistance that develops in four crucial moments [35]: (1) during the transport of cisplatin in the bloodstream, the binding to blood proteins can deactivate the drug itself; (2) because of the inhibitory effect of cisplatin on the CRT1 carrier, there is a decrease in the influx and an increase in the drug efflux; (3) the cytoplasmic inactivation of the drug due to its binding to glutathione (GSH) and excretion catalyzed by glutathione S-transferase enzyme (GST); (4) if the DNA damage repair system, nucleotide excision repair (NER), works properly, then the cancer cell would survive, and cisplatin would not be able to carry out its activity. Oncologists tend to use this chemotherapy in combination with other drugs that act with different mechanisms of action. Combination therapy includes adding Doxorubicin, Gemcitabine, Cyclophosphamide, or Paclitaxel [35,36,37,38].

### 2.2. The Antimetabolite Doxorubicin

The antimetabolite Doxorubicin belongs to the Anthracyclines family, along with Daunorubicin and Epirubicin. This chemotherapy drug is still widely used to treat cancers, including breast, liver, colon cancer, lymphoma, and leukemia [39]. Unfortunately, the significant side effects that occur during and years after treatment will limit its use. Toxicity occurs in specific districts (heart, brain, liver, kidneys), although cardiotoxicity tends to be the most prominent adverse event [40]. Structurally, Doxorubicin is composed of a tetracyclic ring, which gives poor water solubility, two hydroxyl groups, and a sugar attached to the ring with a glycosidic bond, responsible for hydrophilic properties [41]. The mechanism of action of Doxorubicin uses three important pathways simultaneously: the intercalation of DNA, inhibition of topoisomerase II, and formation and accumulation of reactive species [42]. Doxorubicin incorporates between DNA bases, inhibiting DNA polymerase and synthesis, and although Doxorubicin is also capable of binding to RNA and inhibiting RNA polymerase, it has a higher DNA affinity. This intercalation results in double-stranded DNA breaks, fragmented nuclei, condensed chromatin, and induction of apoptosis. In addition, increases in p53 and p21 proteins were also appreciated [43,44]. Topoisomerase II is a cellular enzyme that determines the increase or reduction of DNA misalignment and has a fundamental role during DNA replication, as it favors the breaking and facilitates the separation of the two strands. Doxorubicin is an inhibitor of this enzyme, which makes it impossible to replicate DNA, produces irreparable ruptures, and induces cell death [45,46]. However, the mechanism of action that most induces toxicity is oxidative damage: its quinone group is an electron acceptor in a reaction catalyzed by cytochrome P450 reductase in the presence of NADH dehydrogenase. The reduction of Doxorubicin causes the formation of a semiquinone radical, which induces damage to DNA by cleavage, scission, and degradation [47]. This reaction occurs in the presence of oxygen, favoring the formation of reactive species, such as superoxides, hydroxyl radicals, and peroxides, resulting in further oxidative damage to DNA. In addition, reactive oxygen species can react with iron catalyzed by ferredoxin reductase and can cause further DNA damage and induce lipid peroxidation reactions [48]. Doxorubicin treatment may trigger mitochondrial dysfunction with altered ATP production. Because cardiomyocytes require significant mitochondrial support and a large quantity of energy to function effectively, it is consequential that Doxorubicin treatment determines an important alteration of the cardiomyocytes and the dysfunction of the myocardium [49,50]. The direct consequence is the induction of the autophagic process, which, if it fails to restore the condition of cellular damage, automatically results in an intrinsic or extrinsic pathway of the apoptotic death. Doxorubicin may interfere with calcium homeostasis, compromising the ability of mitochondria to obtain calcium from the cytoplasm. Death of the cardiomyocyte following treatment with Doxorubicin leads to irreversible myocardial damage and left ventricular dysfunction [51]. Cardiotoxicity that is Doxorubicin-induced can occur both during and years after treatment, and it becomes mandatory to prevent this side effect to properly employ these crucial drugs for all patients [52,53]. Several strategies have been tried to reduce cardiotoxicity: (1) limiting the dosage of Doxorubicin; (2) the development of Doxorubicin analog; (3) the use of cardio-protective agents that can be administered in combination with Doxorubicin. The reduction of the Doxorubicin dose (no more than 450 mg/m^2^) was not an excellent option since cancer patients were exposed to cancer treatment with reduced effectiveness, increasing the percent of cancer mortality and inducing the drug resistance [54]. The use of structural analogs of Doxorubicin has been helpless. Epirubicin, the most promising analog of Doxorubicin, produces as severe damages as Doxorubicin when it comes to cardiotoxicity [55]. In Figure 2, the mechanism of action of Cisplatin and Doxorubicin is represented.

### 2.3. The Estrogen-Modulating Agent Tamoxifen

Breast cancer accounts for 23% of all female cancers and often, even today, can lead to death. It is related to a wide variety of risk factors, including genetic predisposition and exposure to estrogen; in particular, prolonged exposure to estrogen increases the incidence of breast cancer through mechanisms dependent on or independent of the receptor. Estrogen can bind to its receptor (ER), and this bond provides a stimulus for the proliferation of mammary gland cells, which increases the risk of developing mutations during DNA replication [56]. The binding of 17-β-Estradiol (E2) to its receptor binds to specific regions of DNA, regulating the expression of many genes involved in vital processes, including development, DNA replication, differentiation, apoptosis, angiogenesis, regulation of the cell cycle, survival, and tumor progression [57]. The hormone E2 can bind to two specific receptors found in the nucleus (ERα and ERβ), where ERβ is more abundant than ERα in healthy human and mouse mammary glands. While, in normal breast tissue, ER β is the dominant receptor, during carcinogenesis, the amount of ER β decreases as the number of ERα increases [58]. Responsive ER breast cancer is often treated with the non-steroidal anti-estrogen Tamoxifen, a selective estrogen receptor modulator with ER agonist/antagonist activities. Tamoxifen competitively binds to ERs and inhibits E2-dependent gene transcription, cell proliferation, and tumor growth [59]. The Tamoxifen-ER complex dimerizes and translocates to the cell nucleus, activating the activation factor 1 (AF1) domain and inhibiting the activation factor 2 (AF2) domain (Figure 3). Then, the complex binds to DNA estrogen response element regulatory regions, but because the ligand-dependent domain AF2 is inactive, the transcription of E2 responsive genes is suppressed [60]. The available data showed that Tamoxifen could reduce the risk of developing ERα-positive breast cancer by 50% both in pre-and post-menopausal women to develop regression of the pathology and the disease in the opposite breast [61]. The mechanism of action of this anti-estrogen agent is based fundamentally on the reduction of cell proliferation by (a) stopping the cell cycle in phase G0/G1 and (b) the down-regulation of some growth factors such as transforming growth factor (TGF) and insulin-like growth factors (IGF-1) [62]. Tamoxifen undergoes oxidative metabolism in the liver and the breast to a lesser extent. Its metabolites are 4-hydroxy-tamoxifen and 4-hydroxy-N-desmethyl-tamoxifen and are formed via the cytochrome P450 route; finally, they become Endoxifen, which shows an affinity from 30 to 100 times greater, respectively, for the estrogen receptor (ERs) compared to Tamoxifen. Endoxifen stabilizes ERβ, promoting its heterodimerization, and increases the inhibitory effects on target gene expression by binding to ERα [63]. Treatment with Tamoxifen involves a wide range of side effects, including endometrial hyperplasia or endometrial cancer, strong hepatotoxic and hepatocarcinogenic effect, hot flashes, venous thromboembolism, night sweats, vaginal dryness, vaginal discharge, depression, sleep changes, weight gain, and decreased sexual functioning [64,65]. Endometrium cancer is one of the most common gynecological neoplasms in the developed world and, as already mentioned, may be a consequence of treatment with Tamoxifen chemotherapy.

### 2.4. The Antimitotic Drug Paclitaxel

Paclitaxel is a very successful anticancer drug, originally isolated from the bark of the Pacific yew tree, Taxus brevifolia, in 1962 and is currently approved for breast, ovarian, and metastatic pancreatic cancers, Kaposi sarcomas, and pulmonary sarcomas [66,67]. Since the demand for Paclitaxel has increased considerably and the gain of Paclitaxel from the bark of the species Taxus is very low (the amount of Paclitaxel extracted from 10 tons of bark or 300 trees can only treat about 500 patients) [68], scientific research has been prompted to look for alternatives to natural sources of the compound. The main alternative pathways were chemical synthesis, chemical semi-synthesis, the pathway of heterologous expression, and the plant cells pathways [69]. The ability of Paclitaxel to impede the dynamics of cellular microtubules is its mode of action (Figure 3). Microtubules are the main component of the cytoskeleton and consist of two similar tubulin polypeptides: the α and the β tubulin dimers. The mitotic spindle, which appears during mitosis, is formed by the continuous assembly and disassembly of microtubules. Therefore, these cytoskeleton components play a significant role in many biological processes of the eukaryotic cell, including cell motility and the correct chromosome segregation [70]. Paclitaxel selectively binds to the β subunit and alters its polymerization and assembly, promoting a dysfunctional synthesis of dynamic spindle formation during cell mitosis. The consequences are the arrest in the G2/M phase of the cell cycle, mitosis suppression, and the subsequent death of cancer cells [71]. In the eukaryotic cells, a spindle assembly checkpoint acts to prevent chromosome missegregation. The mitotic control point delays chromosome separation until each pair has established stable attacks on both mitotic spindle poles. Chromatids connect to the spindle microtubules through their kinetochores. Unaffected kinetochores, which have not established stable attachments to microtubules, activate a cascade of signal transduction that delays mitotic progression by inhibiting anaphase. Paclitaxel promotes unattached kinetochores [72]. This drug also limits tumor angiogenesis and proliferation by acting on genes and cytokines responsible for suppressing cell growth [73]. Finally, it does not affect cancer cell DNA and RNA synthesis or induce DNA molecule damage. Otherwise, this drug promotes the polymerization of tubulins and prevents their depolymerization, stabilizing microtubules [74]. Even though Paclitaxel has a unique mechanism of action and does not alter the structure of biological macromolecules, it causes serious side effects such as myelosuppression, hypersensitivity reactions, neurotoxicity, and cardiotoxicity [75]. Paclitaxel-induced myelosuppression occurs most frequently as leukopenia and/neutropenia, where patients who developed these side effects were 26% and 68%, respectively. Under these circumstances, the fever was a common symptom. These symptoms were exacerbated in individuals receiving radiation or other myelotoxic therapy together with Paclitaxel. However, it is important to note that myelosuppression has a short lifespan in most cases. Thrombocytopenia and anemia may also occur, and these conditions significantly increase the risk of developing cardiotoxicity [76]. In Figure 3, the mechanism of action of Tamoxifen and Paclitaxel is represented.

## 3. Genera *Vaccinium* L., *Citrus* L., *Olea* L., and *Cynara* L.: From Botany to Human Health

The genus *Vaccinium* L. belongs to the *Ericaceae* family of plants, comprising more than 450 species that mainly grow in the northern hemisphere, in countries characterized by cold climates, Central Europe, Russia, and Canada; however, these plants are also present in tropical areas such as Madagascar, Java, and Hawaii [77]. These plants are mostly known for the commercial production of *Vaccinium* berry fruits, and the principal ones come from the species of the sections *Vitis-idaea L* (cranberries), *Oxycoccus* (blueberries), *Cyanococcus* (blueberries), and *Myrtillus* (cranberries) [78].These berries possess high levels of antioxidant compounds (phenolic, flavonoids, tannins), vitamins (A, B1, B2, B3, C), fruits dyes (anthocyanins and carotenoids), and minerals (potassium, calcium, magnesium, phosphorous) [79,80]. Hence, they are widely known for their health benefits and can improve human health conditions, most probably due to the synergistic action of several phenolic biotic compounds present in the berries. In vivo and in vitro studies have shown that berries of the genus *Vaccinium* L. exert antioxidants and anti-inflammatory [81,82] and anticancer [83] activities, but their antiseptic, antimicrobial properties and ability to prevent brain aging and neurodegenerative disorders are also known [84,85]. The antioxidant activity of *Vaccinium* berries is explained by the high phenolic content widely distributed in the leaves, fruits, seeds, and flowers of plants. In particular, blueberries (*Vaccinium Vitis-idaea* L.) present the highest antioxidant activity among berry fruits, including blueberries, blackberries, strawberries, raspberries, and blueberries [86]. Cranberry extract (50 mg/mL), consisting of 5.8% polyphenols, 2.9% flavanols, 1.9% phenolic acids, and 1.5% anthocyanins, showed a significant antioxidant protective effect by reducing hydroxyl radical (OH) and superoxide anion (O^2−^) by 83% and 99%, respectively [87]. The extracts of the *Vaccinium* berries also significantly reduced the expression of NADPH 4 oxidase (NOX4), an oxidizing enzyme with the marked property of producing reactive oxygen species, and increased the levels of the antioxidant enzymes superoxide dismutase 2 (SOD2), glutathione peroxidase (Gpx), and catalase [88]. The compounds with known antioxidant properties in the *Vaccinium* berries are resveratrol, quercetin, luteolin, and rutin [89,90,91]. Since the bioactive compounds, taken with the diet, play their effects proportionally to their bioavailability and can reduce their concentration following gastrointestinal digestion, it is crucial to highlight that the principal genus *Vaccinium* L. berries’ digested compounds maintain their bioactivity [92].

The genus *Citrus* belongs to the *Rutaceae* family and comprises about 160 genera and 1650 species. The primary centers, from which began the diffusion in the other continents of the fruits of the genus citrus, were the tropical and subtropical regions of south-east Asia, of north-east India, of southern China, of the Indo-Chinese peninsula, and of the Malaysian archipelago [93]. The genus *Citrus* species grow preferably in zones where the climate is mild-temperate, with temperatures ranging between 13–30 °C. They hardly tolerate prolonged exposure to the sun, wind, and frost [94]. Hence, the cultivation of the *Citrus* genus expanded significantly towards the north of the Mediterranean area. *Citrus limon* (lemon), *Citrus aurantium* (bitter orange), *Citrus sinensis* (Chinese orange), *Citrus reticulata* (mandarin), *Citrus paradise* (grapefruit), and *Citrus bergamia* are among the most known and used species of the genus *Citrus* [95]. Botanical classification of the genus *Citrus* species is challenging due to the frequent formation of hybrids and the introduction of numerous cultivars through cross-pollination. The purpose of the formation of new hybrids is to obtain fruits with better organoleptic properties or to obtain characteristics that improve their quality, such as taste, juiciness, or absence of seeds [96]. Citrus fruits are abundant sources not only of secondary metabolites (i.e., antioxidant), belonging to the phenolic subclasses (naringenin, naringin, hesperidin, quercetin, rutin, apigenin, phenolic acids, and coumarins) and terpenoids (carotenoids and limonoids), but also of vitamins (A, B1, B2, B3, C, and E) and minerals (Ca, Na, S, Mg, Ni, Fe, Cu, Zn, Mn, Mo, and Se) [97]. This unique chemical profile of the genus *Citrus* is responsible for beneficial health activities, including antioxidant, cardioprotective, antihypertensive, anti-inflammatory, analgesic, lipid-lowing, improving insulin-sensitivity, antidiabetic, antitumor, antimicrobial, and neuroprotective properties [98,99,100,101,102,103,104,105].

The genus *Olea L*. belongs to the family *Oleaceae* and includes around 30 to 40 species. These plants are distributed in the Mediterranean basin, south-east Asia, southern Africa, and eastern Australia. The genus *Olea* includes plants of great economic interest, such as the olive tree (*Olea europaea* subsp. *europaea*). The olive tree is native to the Mediterranean basin and plays a crucial role in nutrition for the fruits (olives) and the extracted oil. For thousands of years, the olive groves have been reliable producers of food and oil, supporting the successive civilizations of the Mediterranean area; the olive tree is also used for its wood. The chemical composition of olive oil indicates that it is rich in bioactive compounds, including vitamins, flavonoids, and polyphenols. The fruit and leaves contain various phenolic compounds such as oleuropein, tyrosol, hydroxytyrosol, cumaric acid, ferulic acid, and caffeic acid [106]. Phenolic compounds have excellent biological properties; among these are antioxidant [107,108], anti-inflammatory and analgesis [109,110], cardioprotective [111,112], anticancer [113], antidiabetic, and neuroprotective effects [114,115].

The genus *Cynara* L. belongs to the *Asteraceae* family, probably native to South America and the Mediterranean area. It is the most numerous plant globally, including 23,000 species and over 1670 genera. The genus *Cynara* L. comprises three botanical varieties: *Cynara cardunculus* var. *scolymus* L. (globe artichoke), *Cynara cardunculus* var. *altilis* (cultivated cardoon), and *Cynara cardunculus* var. *sylvestris* (wild cardoon) [116]. Artichoke is widely used in southern Europe, Portugal, Spain, and Italy. Conversely, the use of wild thistle has evolved as a potential source of solid biofuel/lignocellulosic biomass, biodiesel, seed oil, paper pulp, green fodder, and pharmacologically active compounds [117,118]. In addition, wild artichoke can also act as rennet for cheese production, thanks to the activity of the aspartic proteinases of its flowers [119]. Artichoke is a rich source of several valuable compounds, and phenylpropanoids and sesquiterpenes have valuable biological activities. Numerous scientific studies have chemically characterized the different parts of the plants of *Cynara cardunculus* (in particular, the leaves, the stems, and the flowers) and have identified valuable compounds such as sesquiterpene lactones, including cynaropicrin, as the main constituent [120,121]. It is important to stress that the polyphenols from artichoke leaf extract might also benefit against metabolic and inflammatory-related diseases, including liver, adipose, and cartilage issues [122].

The main botanical features of these genera are described in Table 1.

## 4. Effects of Polyphenols, Contained in Genera *Vaccinium* L., *Citrus* L., *Olea* L., and *Cynara* L, against the Side Effects Induced by Treatment with Selected Anti-Tumor Drugs

When macromolecules, cells, and tissues are exposed to an excess of oxidizing agents and the antioxidant defenses of the cell/organism are insufficient to maintain a redox balance, an oxidative stress state occurs with increased highly reactive unstable chemical species (oxygen and nitrogen free radicals, ROS and RNS, respectively), which can generate metabolic changes, damage, and cell death [138,139]. To date, it is known that oxidative stress is involved in the occurrence of many human pathologies such as metabolic, inflammatory, and neurodegenerative diseases and cancer [140,141,142]. In cancer, on the one hand, the accumulation of oxidative stress is able to contribute, through different mechanisms, to the onset of the disease; on the other hand, the alteration of the redox balance accentuates the proliferation of cancer by the development of chemoresistance [143]. Multiple-drug resistance (MDR) occurs when cancer cells are resistant to chemotherapy drugs and may present as innate or acquired. In innate MDR, cancer cells are already equipped to withstand the anticancer drug used. In contrast, in acquired resistance, cancer cells initially respond to treatment, but subsequently develop extraordinary resistance mechanisms [144]. To date, the development of the acquired resistance to cancer tends to be associated with increased cellular oxidative stress [145]. Processes involved in chemoresistance mediated by oxidative stress include autophagy by endoplasmic reticulum stress, increased progression of the cell cycle, increased number of cancer stem cells, and increased conversion into metastases [146]. In fact, cancer cells that have developed drug resistance have a higher content of reactive oxidizing species than cancer cells that are not resistant to chemotherapeutic agents and are more susceptible to changes in ROS levels. In addition, molecules capable of reducing the generation and accumulation of ROS are potentially useful in the treatment of cancer patients who have developed chemoresistance [147]. To date, it is known that the oxidative stress modulation can be considered a winning strategy to combat MDR, and compounds that reduce cellular ROS levels can increase the death of MDR cancer cells and sensitize MDR cancer cells to certain chemotherapeutic drugs. For this reason, the use of antioxidant compounds of natural origin, such as polyphenols, has become important in order to build a barrier against the generation of reactive oxidant species [148]. The mechanism of action by which polyphenols play antioxidant effects is to be found in the presence of hydroxyl groups linked to the benzene ring that provide the ability to donate a hydrogen atom or electron to free radicals. The consequence is the stabilization of free radicals, which reduces their possibility of damaging cellular components [149].

In order to improve the clinical outcomes of Cisplatin and reduce, at the same time, the side effects caused and the drug resistance, researchers propose the combination of conventional chemotherapy with natural compounds. Ovarian cancer annually causes many deaths worldwide due to the development of resistance to the chemotherapy drugs used. An in vitro study conducted on A2780S and A2780/CP ovarian endometrioid adenocarcinoma cell lines has deepened the effect of oleuropein on cell viability, Cisplatin resistance, and apoptosis. In particular, oleuropein-treated cell lines showed an increase of expression of p21 and p53, while apoptosis inhibitors Bcl-2 and Mcl1 were reduced [150,151]. As a result, oleuropein was able to induce apoptosis, inhibited cell proliferation, and reduced resistance to cisplatin in ovarian cancer cells. At the same time, oleuropein has been shown to reduce Cisplatin-induced oxidative stress and prevent the development of chemotherapeutic complications including hematological tumors. Geyikoğlu et al. demonstrated these results in an in vivo study conducted on male Sprague Dawley rats [152]. Hydroxytyrosol derives from oleuropein, following a hydrolysis reaction, and has proven to be one of the most powerful compounds with antioxidant action. Frequently, the use of the chemotherapy Cisplatin is reduced due to the induction of inflammation and oxidative stress that facilitate the onset of nephrotoxicity [153,154]. Since hydroxytyrosol has anti-inflammatory and antioxidant effects, it has been tested on the kidney of mice following treatment-induced damage with Cisplatin in an in vitro and in vivo study. The results showed that hydroxytyrosol was able to limit Cisplatin-induced inflammation by reducing the NF-κB activation and the TNF-α and IL-1β levels. In addition, hydroxytyrosol decreased the production of malondialdehyde (MDA) and NO increased by Cisplatin [155]. Luteolin belongs to the large group of flavonoid polyphenols and is present in numerous plants, including pinophyte, pteridophyta, peppermint, thyme, and rosemary; it possesses a variety of pharmacological properties such as anti-inflammatory, antimicrobial, antioxidant, antiallergic, and anticancer activities [156]. Wang et al. demonstrated both in vitro and in vivo that luteolin can enhance the therapeutic potential of Cisplatin in ovarian cancer. The treatment with luteolin alone inhibits cell proliferation, but the co-treatment luteolin-Cisplatin further reduces it [157]. Another in vivo study, highlighting the protective role of luteolin against DNA damage and cisplatin-induced oxidative stress, was conducted by Maatouk et al. The results obtained showed that mice receiving luteolin (40 mg/kg), before treatment with Cisplatin (10 mg/Kg), showed a reduction in levels of MDA, catalase, GPx, SOD, and GSH in the liver, kidneys, brain, and spleen, compared to those induced by treatment with Cisplatin alone. In addition, tissue damage, the genotoxic effect, and the side effects generated by the anticancer drug were reduced [158]. Resveratrol is a polyphenolic phytoalexin found in several different plants in nature and, particularly, in grapes. In addition, resveratrol also belongs to phytochemicals with potent antioxidant activity, which provides preferable efficacy with chemotherapeutics but attenuated toxicity in vital tissue [159].

Literature studies have shown that this natural compound possesses consistent antioxidant properties and that it can emphasize the cytotoxic action of Cisplatin in the testicular cancer of rats [160]. An interesting result was provided by a study [161], which highlighted the effect of resveratrol in limiting the toxicity of Cisplatin, especially when higher doses are needed in particularly aggressive tumors such as those of the head and neck. In particular, the co-treatment with Cisplatin-resveratrol on Fadu cells, a hypopharyngeal cell line carcinoma, induced an increase in the effectiveness of Cisplatin, thanks to an increase in apoptotic death, accompanied by the blocking of the cell cycle in the G0/G1 phase. In addition, resveratrol has reduced the harmful effects induced by Cisplatin. A recent in vitro study on SKOV3 cells, isolated from a female with ovarian adenocarcinoma, showed that treatment with naringin, at different concentrations and for different times, was able to reduce Cisplatin resistance [162].

To date, the use of plant polyphenols is exponentially increasing and is considered an adjuvant therapy in combination with Doxorubicin. Polyphenols have an interesting antioxidant property, and these compounds can prevent tumorigenicity, thanks to their anti-proliferative and cardio-protective effects [163]. Resveratrol, found in large quantities in red wine, berries, and grapes’ skin, can reverse the Doxorubicin-induced cardiotoxicity by reducing oxidative stress and diminishing lipid peroxidation. Polyphenol employment reduces malondialdehyde level and protects the enzyme superoxide dismutase activity [164]. Resveratrol prevents the apoptotic death of cardiomyocytes induced by Doxorubicin, as demonstrated by the reduced expression of the protein p53 and Bax [165] and autophagy dysregulation [166]. Finally, resveratrol can be used in conjunction with Doxorubicin treatment to improve the therapeutic efficiency of chemotherapy while also protecting the heart from cardiotoxicity [167]. Apigenin is a natural flavonoid found in fruits and vegetables with antioxidant, anti-inflammatory, anticancer, and cardioprotective properties [168,169]. Apigenin is a good iron chelator and scavenger of free radicals. It can be used synergistically with Doxorubicin to treat leukemia, causing cell cycle arrest and inducing apoptotic death. In addition, apigenin attenuates Doxorubicin-induced cardiotoxicity by activating a PI3K/Akt/mTOR-dependent pathway and has beneficial action in many cancer models without developing toxicity. Finally, this compound can overcome the chemoresistance to the Doxorubicin, reducing the expression of Nrf2, notoriously involved as a transcriptional regulator of drug resistance [170,171]. The polyphenolic fraction of bergamot (BPF) comes from the pressing of the fruit of the bergamot plant (Citrus bergamia), a plant endemic to the Calabria region (Italy), with a flavonoid and unique glycosidic profile. Carresi and colleagues demonstrated, in an in vivo model of Doxorubicin-induced heart damage, the antioxidant and cardioprotective effects of BPF. The reduction of heart function occurs after 21 days of Doxorubicin treatment, and this adverse effect has been significantly attenuated when animals were co-treated with BPF. BPF’s protective effect has also been linked to the activation of an autophagic protective mechanism. [172]. Numerous other polyphenols protect against cardiotoxicity induced by treatment with Doxorubicin, and the use of this drug combined with natural compounds, which increase the death of cancer cells while protecting the heart from cardiotoxicity, is highly recommended [173]. An important and well-organized study by Tavga [174] showed, in an in vivo model, that quercetin was able to induce protection against Doxorubicin-induced cardiotoxicity in rats. The effects were measured on blood samples and levels of troponin, creatine, phosphokinase, C-reactive protein, total antioxidant capacity, lactate dehydrogenase, and total lipid profile were evaluated. The results showed that treatment with Doxorubicin produced a significant increase in the level of troponin, total cholesterol, CRP, low-density lipoprotein, LDH, triglycerides, and atherogenic index of plasma and that quercetin, co-treated with Doxorubicin, reverted these effects, reporting values in a normal range. Histopathological findings were also provided which supported biochemical findings. The cardioprotective effects of quercetin could be attributed to the effects of antioxidant, anti-inflammatory, hypolipidemic and antiatherogenic activity, indicating this natural compound as an excellent therapeutic candidate to be tested in the clinical field.

Luminal breast cancer is an aggressive disease that is resistant to chemotherapy. Numerous studies have shown that oleuropein and olive oil polyphenols exert a strong antitumor activity in multiple human cancers [175,176,177]. For this reason, it was interesting to examine the effect of the combination of oleuropein with tamoxifen chemotherapy. In particular, an in vitro study showed that treatment with oleuropein inhibited the growth of human breast cancer cells BT-474, MCF-7, and T-47D and that the combined treatment oleuropein-tamoxifen led to a synergistic inhibition of the growth of the same cell lines [178]. Recently it has been shown that the flavonoids contained in fruits belonging to the genus *Citrus* (*Citrus reticulata* and *Citrus aurantiifolia*) have anticancer, antiproliferative, and estrogenic effects [179]. In particular, hesperidin, hesperetin, naringenin, tangeretin, nobiletin, and naringin have inhibiting activity on the growth of some cancer cells through various mechanisms. It has also been shown that these compounds play a synergistic effect with several chemotherapy agents [180]. The mechanisms of action involved include apoptosis, cell cycle modulation, and antiangiogenic effects. Finally, the natural compounds mentioned, in combination with tamoxifen, have demonstrated a synergistic effect on the cell line of human breast cancer MCF-7 [181]. Tamoxifen leads to a strong hepatotoxic and hepatocarcinogenic effect in rats when used at doses comparable to human therapeutic doses. As already mentioned, this drug is metabolized in the liver by the cytochrome P450 family; since there is increasing evidence of the hepatoprotective role of flavonoids, it would be possible to treat breast cancer with Tamoxifen together with these hepatoprotective plant extracts in order to reduce the suffering liver [182,183]. Another plant extract with known hepatoprotective properties is the cynaropicrin, a sesquiterpene lactone extracted from the artichoke plants and the most biologically important class of secondary metabolites of the plants [184,185,186,187,188]. Although no specific studies have been carried out on the use of Tamoxifen and cynaropicrin, it would be interesting to consider the employment of this polyphenol as an adjuvant drug in chemotherapy that induces liver impairment.

The administration of the polyphenol resveratrol (4 mg/kg in 40 mL normal saline, in rabbits) combined with Paclitaxel significantly decreases myelosuppression’s degree and duration [189]. The hypersensitivity reactions associated with Paclitaxel include dyspnea, with or without bronchospasm (81%), hives, redness, or erythematous rash (74%), hypotension (41%), and angioedema (18.5%). These side effects occur within the first hour of Paclitaxel infusion and are observed despite premedication with antihistamines and corticosteroids [190]. The concomitant use of certain polyphenols is crucial to preventing or treating this symptomatology. Some polyphenols, such as quercetin, resveratrol, oleuropein, silibinin, and many tannins and carotenoids, exhibit antiallergic effects, including the inhibition of histamine release, reduction of proinflammatory cytokines, and leukocyte production [191]. The mechanisms shown by these natural compounds are that: (a) polyphenols can influence the formation of the allergenic-IgE complex, and (b) polyphenols may delay the binding of this complex to its receptors [192]. Treatment with Paclitaxel can also generate hepatotoxicity; in fact, a clinical study conducted on 402 patients showed elevated bilirubin values and transaminases dysfunction [193]. Recently, an in vitro study of human breast cell lines (MCF7 and MDA-MB231) highlighted that treatment with a polyphenolic fraction of artichoke extract, together with Paclitaxel, managed not only to increase the anti-proliferative effect of the chemotherapy drug but also, at appropriate concentrations, to protect liver cells from Paclitaxel-induced hepatic toxicity [194]. Many chemotherapeutic agents, including Paclitaxel, cause chemotherapy-induced peripheral neuropathy (CIPN), leading to treatment suspension, altering the patient’s quality of life and reducing the survival rate. Mechanical and thermal hypersensitivity accompany CIPN and resolves within weeks, months, or years of drug discontinuation. Because the etiology of CIPN has not been fully explained, there is currently no available preventive strategy or effective treatment. However, considerable evidence reveals that free radicals play a role in many neurodegenerative diseases, and recent research has demonstrated the importance of oxidative stress in developing CIPN targeting the overproduction of peroxynitrite for the prevention and reversal of Paclitaxel-induced neuropathic pain [195,196,197]. The bergamot polyphenolic fraction (BPF), a natural derivative antioxidant, can play a crucial role in reducing CIPN. Recent data show that Paclitaxel administration causes mechanical allodynia and thermal hyperalgesia, starting on day seven and ending on day fifteen. In addition, Paclitaxel-induced neuropathic pain is correlated to protein nitration in the spinal cord, including MnSOD, glutamine synthetase, and the glutamate transporter GLT-1 [198]. Furthermore, some immune system cells, such as mast cells and basophils, seem to be directly involved in CIPN [199,200]. For example, histamine release from mast cells plays a fundamental role in developing thermal hyperalgesia and mechanical allodynia in mice [201]. Furthermore, as mast cells are adjacent to sensory nerves, histamine is released, and neurons participate in spinal nociceptive transmission by releasing neuromediators of pain [202]. Since quercetin, one of the polyphenolic flavonoids distributed in various plants and with biological activity, works specifically as a stabilizer of mast cells and inhibits histamine release, it has been shown that quercetin attenuates Paclitaxel-induced neuropathic pain [203]. In Figure 4, the chemical structure of the selected chemotherapeutics and the natural compounds contained in the genera considered are represented.

## 5. Discussion and Conclusions

Cancer is caused by multiple molecular alterations that lead to the loss of cell growth control and differentiation, resulting in uncontrolled hyperproliferation that, if not stopped, ends with tumor formation [204,205]. The treatment of cancer with chemotherapy drugs highlights significant limits: on the one hand, the numerous systemic side effects with severe toxicity to normal tissues and, on the other hand, the development of drug resistance, which makes treatment even more difficult [206,207]. In fact, due to often disabling chemotherapy-induced side effects, these drugs cannot be administered at doses where they could potentially eradicate all cancer cells [208]. This effect is partly responsible for the selection of resistant cancer cells that no longer respond to the drug’s effect and allow the growth of tumors refractory to the further cancer treatment. Toxicity and drug resistance are two sides of the same coin that significantly reduce the efficacy of chemotherapy responses [209]. Cisplatin, for example, one of the most efficacious and first metal-based chemotherapeutic drugs, does not show its maximum potential due to significant induced side effects and the development of drug resistance. The primary toxicities derived from Cisplatin therapy are nephrotoxicity, hepatotoxicity, ototoxicity, neurotoxicity, and gastrointestinal toxicity [210]. Resistance to Cisplatin depends on multiple factors, including an insufficient concentration used, inactivation of the active ingredient due to its binding to different proteins, dysfunction of proteins intended to induce apoptotic death, and the increase in DNA repair processes [211]. The toxicity induced by Doxorubicin mainly affects cardiac function and is consequent to excessive oxidative stress induced by the drug, which favors redox modifications of the biological macromolecules. In particular, there are reactions of protein carbonylation and lipid peroxidation in the heart muscle and modifications of myofibril proteins, such as troponin I, tropomyosin, and actin, that compromise cardiac function [212]. Despite the remarkable clinical success of Tamoxifen therapy, patients treated with this chemotherapy showed the induction of mutations in the genes of cytochrome P450 enzymes (CYP), a superfamily of the proteins responsible for drug metabolism and their detoxification, with reduced effectiveness of the anti-antitumoral drug. In addition, therapeutic resistance develops after receiving hormonal therapy due to various mechanisms, including increased mitogen-activated signaling kinase protein (MAPK) and abnormalities in the estrogen receptor gene [213,214]. Finally, the effectiveness of Paclitaxel is also limited by various side effects associated with its use, and the main ones are hypersensitivity and neuropathies. Hypersensitivity includes bronchospasm, hot flashes, wheezing, hives, erythematous rash, hypotension, angioedema, chest pain, abdominal pain, fever, or rigors, and some of these symptoms are already visible within the first ten minutes of administration [215]. Treatment with Paclitaxel can generate different mechanisms of drug resistance. The overexpression of efflux proteins facilitates the extrusion of the drug in advance; in addition, Paclitaxel may result in reduced activity of cyclin-dependent kinase-1, which induces drug resistance. Finally, changes in the expression of miRNAs may occur, who play a crucial role in chemotherapy in many therapeutic approaches [216,217]. The use of several Polyphenols, together with anticancer drugs, has been shown to reduce chemotherapy’s side effects and improve the effectiveness of drugs. Many compounds of natural origin, especially polyphenols, have the ability to act synergistically with anticancer drugs, improving their effectiveness [218,219]. In particular, polyphenols, taken at the correct concentrations for certain times, are able to reduce the viability of cancer cells by increasing apoptotic death, inhibiting the continuation of the cell cycle, and modulating gene expression [220,221]. The concentration and times used for polyphenols are fundamental; while high concentrations and chronic treatment times tend to have harmful effects, reduced concentrations and short treatment times tend to be protective [222]. The reduction of side effects is linked to the ability of these drugs to perform an antioxidant action. In fact, if oxidative stress was present, it would contribute to aggravate an already particularly compromised situation. Reducing the accumulation of reactive oxygen species could, in fact, protect the areas of the body affected by cancer [223,224]. Another mechanism exerted by polyphenols, to relieve the side effects of anticancer drugs, is the extinguishing of the inflammatory process. In fact, frequently, cancer or the use of antitumoral drugs can promote an inflammatory effect at the systemic level [225,226,227]. Polyphenols are able to reduce the expression of many cytokines and inflammatory molecules, which are responsible for igniting the inflammatory process [228,229,230]. Chemotherapy has a limited use due to the induction of drug-resistance, which makes the tumor mass not responsive to drug treatment, as previously explained [231,232,233,234,235,236]. Recent in vitro studies have shown that polyphenols are able to overcome drug-resistance in different types of cancer, including breast, lung, prostate, and colon, using the following mechanisms: (1) inhibit efflux pumps that extrude anticancer drugs; (2) increase the level of drug absorption; (3) increase cell apoptosis; (4) decrease the proliferation of cancer stem cells [237,238].

The intake of polyphenols leads to numerous benefits, but questions remain unresolved, such as “What is the amount of these natural compounds to be consumed to ensure protective effects on health?” and “The exceedance of these quantities may rise toxic effects?”. Correct and unequivocal answers are challenging to provide since there are multiple variables related to the plant product (degree of ripeness, soil composition, geographical latitude, adverse conditions for the plant), chemical composition, intestinal absorption, metabolism, and conjugation reactions. In general, the intake of polyphenols within the commonly consumed food range does not cause any toxic damage. When intake levels are increased but remain within reasonable limits, no intolerable harm occurs; finally, when intake becomes much higher, and the consumption of food supplements and pharmacological approaches is necessary, undesirable effects are observed [239,240]. When the effect of polyphenols is related to cancer, it is necessary to use concentrations that can stop uncontrolled cell proliferation and maintain the antioxidant and anti-inflammatory effect that do not complicate an already compromised condition. For this reason, it is crucial to know the mechanisms of action of the known anticancer drugs, their side effects, and how polyphenols can help reduce and/or revert these alterations [241,242,243,244,245,246,247]. The use of polyphenols from the genera *Vaccinium* L., *Citrus* L., *Olea* L., and *Cynara* L. are recognized for their synergistic effects with numerous anticancer drugs and their ability to alleviate the side effects from the use of such drugs. Further studies must be carried out to verify the potential beneficial effects of polyphenols at the preclinical and clinical levels and apply and supplement them to traditional chemotherapy. The use as adjuvants of polyphenols in the clinical anticancer routine would be a significant step forward in treating cancer and improving patient quality of life.

## Figures and Tables

**Figure 1 nutrients-14-01574-f001:**
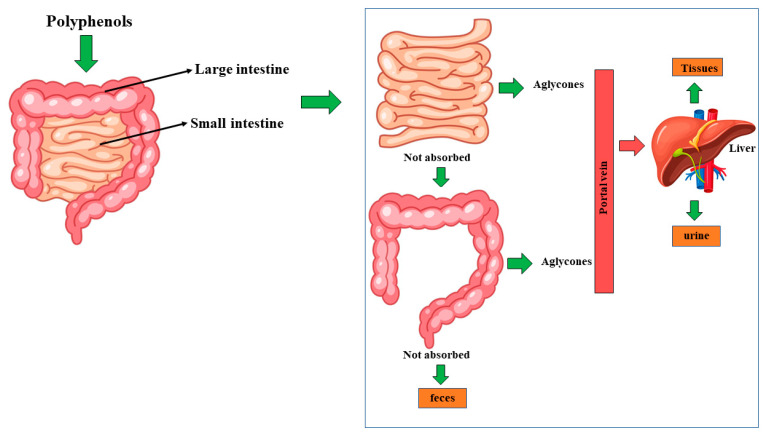
Absorption and metabolism for dietary polyphenols.

**Figure 2 nutrients-14-01574-f002:**
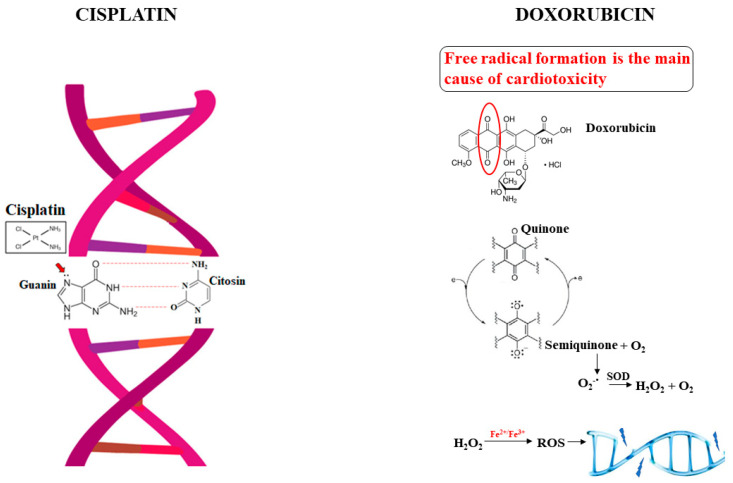
Mechanism of action of Cisplatin and Doxorubicin.

**Figure 3 nutrients-14-01574-f003:**
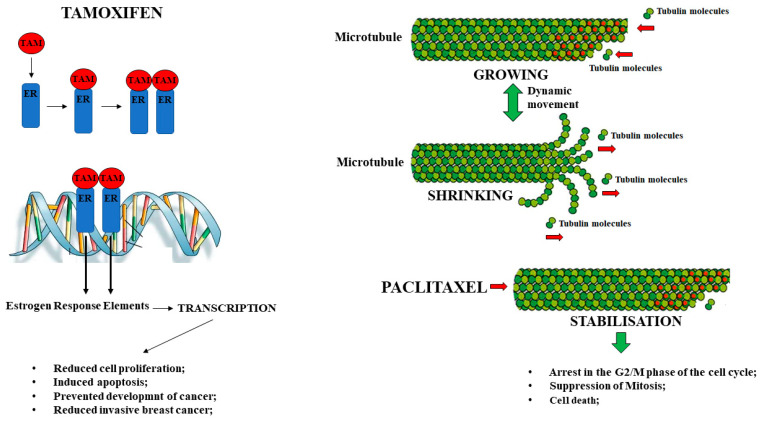
Mechanism of action of Tamoxifen and Paclitaxel.

**Figure 4 nutrients-14-01574-f004:**
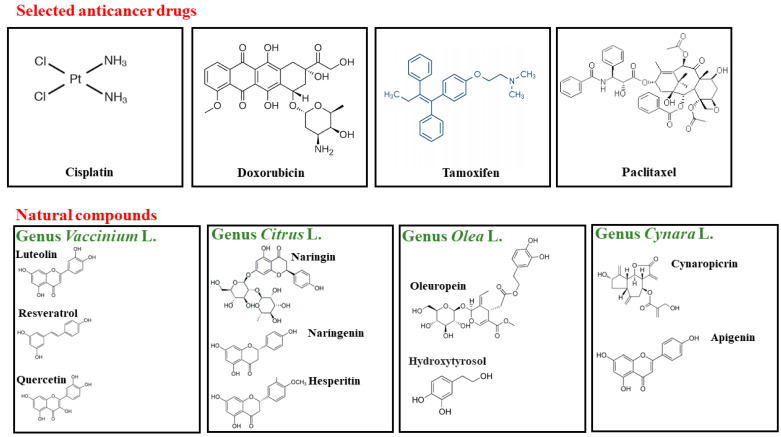
Chemical structure of selected anticancer drugs and natural compounds contained in the genera considered.

**Table 1 nutrients-14-01574-t001:** Botanical features of genera *Vaccinium* L., *Citrus* L., *Olea* L., and *Cynara* L.

Botanical Characteristics	Genus *Vaccinium* L. [81]	Genus *Citrus* L. [123]	Genus *Olea* L. [124]	Genus *Cynara* L. [125,126,127,128,129,130,131,132,133,134,135,136,137]
Trunk/shrubs	Almost always small shrubs of modest size or even creeping.	Shrubs or evergreen trees with a height varying from 3 to 15 m.	Small evergreen trees, from 12 to 20 ft. high; Rigid branches and a grayish bark.	Herbaceous forms with height between 50 and 250 cm. The stems are erect, branched, and robust.
Leaf	Leathery, oval, and evergreen.	Ovoid or elliptic; coriaceous.	Opposite, evergreen, petiolate, and coriaceous.	Basal and cauline leaves; the lamina is pinnatisect and very thorny.
Flowers	United in clusters and terminal; white-pink; petals welded.	White or reddish; grows individually in leaf axils; consists of five petals.	Agglomerate, fasciculate, racemose, or decussate type with terminal or axillary posture. Hermaphrodite, actinomorphic, and tetracyclic.	Vast and globose terminal heads. Tubulose (actinomorphic), hermaphrodite, and fertile. Color is pink, purple, or violet.
Fruit	False fleshy berries of small and medium size.	Modified berry known as “hesperidium”.	Drupe, ovoid, globose, or oblong; mesocarp is fleshy; endocarp is hard.	Achenes with pappus. The shape of the achene is cylindrical and slightly angular. The pappus is formed of long deciduous or persistent feathery bristles.

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
