# Peer review of "The Employment of Genera Vaccinium, Citrus, Olea, and Cynara Polyphenols for the Reduction of Selected Anti-Cancer Drug Side Effects"

_nutrients, 2022, doi:10.3390/nu14081574_

Round 1

Reviewer 1 Report

This manuscript has a great deal of information and references. Individual sections are well-written. However, the separate sections seem disparate, and the overall impact of the manuscript is thus not as clearly apparent as it could be.  Additional text to provide framing, summaries, and transitions would increase ease of reading.

Line 65-67  Three genera are described- why were these 3 chosen? Are they the most common, or most studied, or most accessible as food, or most antioxidant-rich? Is there previous data linking these foods to cancer prevention? There is some description of the reasoning within each plant’s description (see 147-150), but it is overshadowed by a wealth of botanical detail that seem unnecessary given the scope of the paper. Consider use of pictures rather than verbal descriptions.

Similarly, why were cisplatin, doxorubicin, tamoxifen and paclitaxel chosen? Do they represent the different mechanisms described in 222-238? Explanation added to the end of the section “2. Cancer pharmacological treatments …” would make this clearer.

Mechanisms of action of the drugs are described in detail, but do tend to obscure the discussion of protective plant components. It is most helpful when the individual antioxidant component is explicitly linked to the relevant genus, for example artichoke component cynaropicrin and bergamot polyphenolic fraction (line 384-393, 450-456).

Rather than diagrams of drug mechanism of action, could there be a diagram of possible beneficial targets of plant components?

Curcumin (281-299) is described as a prototype example of antioxidant effect, but isn’t one of the chosen genera. However, there appears to be more information on it than on Olea components.

The manuscript describes the role of antioxidants, etc., in mitigating effects of cancer treatment. What about the possibility that there could be impairment of chemotherapy mediated by free radical mechanisms?

The conclusion should include a summary describing the benefits of plant components and reiterating the value of the three genera as foods or supplement sources in cancer treatment- if the paper is about these plants they should be featured more prominantly in the conclusions.

Author Response

Dear reviewer, you will find attached a word document with my responses to your suggestions.

Reviewer 2 Report

The paper entitled ‘The employment of genera Vaccinium, Citrus, Olea and Cynara polyphenols for the reduction of anti-cancer drugs side effects’ aimed to characterization of selected plant genus against anti-cancer drugs side effects. The subject of the review is interesting nevertheless there are numerous points which should be improved. Please consider the following suggestions:

  1. Abstract: too long, please reduce
  2. Keywords: should be without the word ‘genus’
  3. Introduction:
  • The point 1.1 should be a separate paragraph with subsections.
  • All family names should be in italic
  • Lines 80-82: reference needed
  • Line 94: please improve signs of radicals
  • Lines: 130-142: references needed
  • Lines 151-153: references needed
  • Lines 157-175: references needed
  1. Cancer: pharmacological treatments and their side effects. The use of the natural compounds: Please inform whether the studies were performed in animal model or human. The introduction to the section should be divided into paragraph because the present form is hard to read. Additionally, please provide more detail information about all presented study results.
  2. Please provide structures of all presented active substances (i.e. cisplatin)
  3. All sections describing interactions the active compound with polyphenols should be divided in a few paragraphs. In present form it is hard to read.
  4. All figures should be provided close to their description in main text, not at the end of section.
  5. Figure 2: there should be iron ion instead of atom iron
  6. Figure 4. Does not bring new information. In my opinion can be delete.
  7. Conclusions: should not include references

Author Response

Dear reviewer, you will find attached a word document with my responses to your suggestions

Reviewer 3 Report

The review article presents the employment of genera Vaccinium, Citrus, Olea and Cynara  polyphenols for the reduction of anti-cancer drugs side effects. The manuscript is suitable for Nutrients, yet below points need to be further explained or revised:

  • This article focuses on several anticancer drugs, so I suggest adding the word "selected" in the title (“…selected anti-cancer drugs….”).
  • In general, the review concerns old anti-cancer drugs that have been used for many years. In my opinion the authors should be tempted to expand on the article with information on the effects of polyphenols on side effects in relation to new anti-cancer drugs.
  • Introduction is too long. In my opinion, the subsection 1. Genera Vaccinium L., Citrus L., Olea L. and Cynara L.: from botany to human health should be a separate section.
  • Discussion is poor and should be enriched with comparison of the obtained results with other references.
  • The Authors should familiarize themselves with the proper format for References and make appropriate corrections (g. references 9, 49, 50, 204 ... - please correct the font, reference 12 - Vaccinium vitis-idaea L. should be in italics).

Author Response

(The authors gave the same response as above.)

Round 2

Reviewer 2 Report

The paper has been improved in accordance with reviewer suggestions. The paper is suitable for publication. 

Reviewer 3 Report

The authors corrected the work according to the reviewer's guidelines. I have no objections.